# PMMA Bone Cements Modified with Silane-Treated and PMMA-Grafted Hydroxyapatite Nanocrystals: Preparation and Characterization

**DOI:** 10.3390/polym13223860

**Published:** 2021-11-09

**Authors:** Do Quang Tham, Mai Duc Huynh, Nguyen Thi Dieu Linh, Do Thi Cam Van, Do Van Cong, Nguyen Thi Kim Dung, Nguyen Thi Thu Trang, Pham Van Lam, Thai Hoang, Tran Dai Lam

**Affiliations:** 1Institute for Tropical Technology, Vietnam Academy of Science and Technology (VAST), 18 Hoang Quoc Viet, Cau Giay, Hanoi 10000, Vietnam; mdhuynh@itt.vast.vn (M.D.H.); dvcong@itt.vast.vn (D.V.C.); trangktnd@itt.vast.vn (N.T.T.T.); hoangth@itt.vast.vn (T.H.); trandailam@gmail.com (T.D.L.); 2Graduate University of Science and Technology, Vietnam Academy of Science and Technology (VAST), 18 Hoang Quoc Viet, Cau Giay, Hanoi 10000, Vietnam; nguyendieulinhcnh@gmail.com; 3Hanoi University of Industry, 298 Cau Dien, Bac Tu Liem, Hanoi 10000, Vietnam; docamvan85@haui.edu.vn; 4National Academy of Education Management, 31 Phan Dinh Giot, Thanh Xuan, Hanoi 10000, Vietnam; dungntk@niem.edu.vn; 5Institute of Chemistry, Vietnam Academy of Science and Technology (VAST), 18 Hoang Quoc Viet, Cau Giay, Hanoi 10000, Vietnam; lamvcvh@yahoo.com

**Keywords:** HAP-modified bone cement, vinyltrimethoxysilane-treated HAP, PMMA-grafted HAP, bending, ISO 5833

## Abstract

In this study, vinyltrimethoxysilane-treated hydroxyapatite (vHAP) and PMMA-grafted HAP (gHAP) were successfully prepared from original HAP (oHAP). Three kinds of HAP (oHAP, vHAP and g HAP) were used as additives for the preparation of three groups of HAP-modified PMMA bone cements (oHAP-BC, vHAP-BC and gHAP-BC). The setting, bending and compression properties of the bone cements were conducted according to ISO 5833:2002. The obtained results showed that the maximum temperature while curing the HAP-modified bone cements (HAP-BCs) decreased from 64.9 to 60.8 °C and the setting time increased from 8.1 to 14.0 min, respectively, with increasing HAP loading from 0 to 15 wt.%. The vHAP-BC and gHAP-BC groups exhibited higher mechanical properties than the required values in ISO 5833. Electron microscopy images showed that the vHAP and gHAP nanoparticles were dispersed better in the polymerized PMMA matrix than the oHAP nanoparticles. FTIR analysis indicated the polar interaction between the PO_4_ groups of the HAP nanoparticles and the ester groups of the polymerized PMMA matrix. Thermal gravimetric analysis indicated that mixtures of ZrO_2_/HAPs were not able to significantly improve the thermal stability of the HAP-BCs. DSC diagrams showed that the incorporation of gHAP to PMMA bone cement with loadings lower than 10 wt.% can increase T_g_ by about 2.4 °C.

## 1. Introduction

Acrylic bone cements, or polymethylmethacrylate (PMMA) bone cement, have been widely used in orthopedics and trauma surgery, such as in artificial joint replacement, the treatment of bone defects or osteoporosis and vertebroplasty [1,2,3]. Nevertheless, similar to most biomaterials, PMMA bone cement has its own drawbacks, such as its high exothermic temperature while curing, low bioactivity or bioinertia and toxicity of the residual monomer [4,5,6]. Therefore, numerous studies reported one or several of the drawbacks of acrylic bone cements [7]. In order to improve the bioactivity, biocompatibility, osteointegration ability and some other properties, bioactive additives are often used to modify PMMA bone cement and to develop new type, called bioactive acrylic bone cement [8].

Hydroxyapatite (HAP) is a compound with a molecular formula of Ca_10_(PO_4_)_6_(OH)_2_, which appears almost in milk-white in color. It is present in crystalline in rod-like, needle-like, layer-like, and sphere-shaped crystals/nanocrystals. The crystalline structures of HAP have hexagonal and monoclinic forms [9]. Due to a great similarity to biological apatite, HAP is used for a variety biomedical applications; furthermore, it was introduced into acrylic bone cement as a bioactive additive [10,11,12,13,14,15,16,17,18,19]. In general, it is desirable to incorporate as high an amount of HAP as possible to improve bioactivity, with a lower reduction in other required properties [20]. It was reported that HAP can influence the mechanical properties of PMMA bone cement, and the results are different in reported studies [16,21,22,23,24]. However, the enhancement of mechanical properties of polymer nanocomposites can usually be achieved at low additive loadings, e.g., in the range of 1–5 wt.%, or somewhat higher, only in the case of a strong interaction between the additives and the polymer matrix [25,26]. Therefore, in order to enhance the compatibility and interaction between HAP and polymer, most of the aforementioned studies used an organic functionalization approach, in which the surfaces of HAP particles were modified with silane coupling agents or other organic compounds, or even polymers. To our best knowledge, little attention was focused on applying polymer-grafted hydroxyapatite hybrids, which are synthesized by grafting polymer onto HAP nanoparticles and using an extraction process to remove homopolymers and obtain a grafted hybrid material. Liu Q. et al. showed that the surface hydroxyl groups of hydroxyapatite have the ability to react with organic isocyanate groups, from which some acrylic monomers, such as methyl methacrylate (MMA), nbutyl methacrylate (BMA), and hydroxyethyl methacrylate (HEMA) can be polymerized onto the surfaces of HAP particles. Thermogravimetric analysis confirmed the presence of grafted polymers on the surfaces of HAP powder particles (20–26 wt.%). As demonstrated by the authors, the stronger grafted composites can be applied in bone cements and dental materials [27]. Goranova K.L. et al. prepared hydroxyapatite-poly(d,l-lactide) nanografts by using “graft-from” polymerization technique towards their application as additives in bone cements. Their study demonstrated that the surface hydroxyl groups of HAP particles can be used as initiators in the ring-opening polymerization of d,l-lactide, resulting in the efficient and reproducible growth of poly(d,l-lactide) chains onto the surface of HAP [28].

In this study, a vinyltrimethoxy silane (VTMS) coupling agent was introduced to the surfaces of HAP nanocrystals (or nanoparticles) to increase their organic affinity and compatibility with the polymer matrix. It was expected that the vinyl groups of vHAP would be copolymerized with metylmethacrylate (MMA, the main component of liquid part) in the solidification of bone cement for the enhancement of its mechanical properties. The copolymer product is PMMA-grafted HAP (gHAP), which was present in the cured bone cement. When preparing the powder part of bone cement, the aforementioned VTMS-treated HAP (vHAP) was mixed with other solid components, including benzoyl peroxide (BPO), which can cause the polymerization of the vinyl groups of vHAP when stored for a long time, which leads to the loss of the activity of vHAP nanocrystals. In order to overcome the problem of the instability of vHAP, the pre-prepared gHAP was synthesized by using common free radical polymerization between vHAP and MMA. It should be noted that when using this technique, the PMMA/vHAP nanocomposite product consists of PMMA homopolymer and gHAP. For the purpose of using HAP as a biocompatible component for bone cement, PMMA homopolymer with low molecular weight should be removed; therefore, an extraction process is needed to collect gHAP from PMMA/vHAP nanocomposites. The product of gHAP obtained through this approach is more stable in the presence of BPO than vHAP.

Following the above procedure, we used PMMA-grafted HAP (gHAP), VTMS-treated HAP (vHAP) and origin HAP (oHAP) as additives for PMMA bone cements. The effects of the gHAP, vHAP and oHAP on the mechanical properties, setting, and mechanical properties of the HAP-modified PMMA bone cements were investigated based on ISO 5833 and related standards for acrylic bone cements. Furthermore, Fourier transform infrared (FTIR) spectra, field emission electron microscope (FESEM), thermal gravimetric analysis (TGA), differential scanning calorimetry (DSC) of oHAP, vHAP, gHAP as well as PMMA bone cements modified with these HAPs were investigated and are discussed in this paper.

## 2. Materials and Methods

### 2.1. Materials

The poly(methyl methacrylate) (PMMA, average molecular weight of 120,000 Da), monomethyl ether hydroquinone (MeQH, 98%), methyl methacrylate (MMA, 99%, containing ~30 ppm MeHQ) and vinyl trimethoxy silane (VTMS, 98%) were purchased from Sigma-Aldrich (St. Louis, MO, USA). Fine PMMA powder was prepared from the original PMMA by using a mini-high-speed pulverizing machine (model VNS-800A, Vinastar, Hanoi, Vietnam) and sifting with a 250 micron-aperture sieve, dried and stored in a glass bottle. Due to the friction of the mesh and the manual sifting, the obtained bead sizes of the PMMA powder were in the range of 10–110 µm in random shapes.

The nanocrystalline calcium hydroxyapatite (HAP, Ca_10_(PO_4_)_6_(OH)_2_, 96.9%), Ca/P molar ratio of 1.67, particle size of 30–80 nm, specific surface area of 110 cm^2^/g, (Ca, P, CO_3_^2−^ and HPO_4_^2−^ contents of 39.51, 18.38, 0.021 and 0.603%, respectively) was kindly provided by Laboratory of Inorganic Chemistry, Institute of Chemistry, Vietnam Academy of Science and Technology. This product was previously synthesized through the precipitation method, using extra pure Ca(OH)_2_ and H_3_PO_4_ as raw materials at room temperature, and finally annealed at 1000 °C [29].

Nano zirconia (ZrO_2_, 99.9%) with an average particle size of 40 nm was purchased from the Aladdin Chemical Corporation (Shanghai, China). Benzoyl peroxide (BPO: 75% BPO, 25% H_2_O), methanol (MeOH, 99.7%), acetone (99.7%), 1,4-dioxane (dioxane, 99%), tetrahydrofuran (THF, 99.5%) and α,α’-azobis(isobutyronitrile) (AIBN, 98%) were the reagent grade products of Guangzhou Chemical Regents Company, Ltd., (Guangzhou, China). Dioxane was purified by passing it through a column filled with dried silica gel. BPO was dried in a vacuum oven at room temperature for 48 h and stored in a glass bottle before use.

### 2.2. Sample Preparation

#### 2.2.1. Surface Treatment of HAP with VTMS (vHAP)

To a 250 mL Erlenmeyer flask, 20.0 g (20 mmol) of original HAP (oHAP) powder was wetted with VTMS solution, which comprised VTMS (3 g, 20 mmol), 15 mL methanol and H_2_O (3.7 g, (VTMS:H_2_O:HAP molecular ratio is of 1:10:1). The mixture was mixed with a spatula while adding 2 g ammonia solution, 25%. The flask was closed with a rubber stopper and incubated at 40 °C for 48 h. The wet powder was recovered by several cycles of washing with acetone and centrifuging at 6000 rpm for 6 min in order to remove residual VTMS. The VTMS-modified HAP (labeled as vHAP) was dried to a constant weight in a vacuum oven at 40 °C and ground by an agate pestle-mortar set.

#### 2.2.2. Synthesis of PMMA/vHAP Nanocomposites and PMMA-Grafted HAP (gHAP)

The PMMA/vHAP nanocomposite was synthesized by using the in situ polymerization method and the homopolymer extraction process was performed to obtain a sole PMMA-g-HAP hybrid. Before use, the MMA was passed through a column filled with basic alumina for removing the MEHQ inhibitor. Into a 250 mL double-neck round bottom flask equipped with an oval coated-magnet were added the vHAP (10 g), pure MMA (10 g), AIBN (0.10 g), dioxane (40 mL) as a solvent of PMMA and MeOH (40 mL) as a nonsolvent of PMMA. The flask was closed with a rubber septum to allow the bubbling process with N_2_ gas at room temperature for at least 20 min by using a long and a short needle inserted through the septum. The flask was then replaced in a preheated oil bath at 60 °C on a magnetic hot plate stirrer; the polymer/inorganic hybridization reaction was allowed to proceed for 6 h at 60 °C. A monomer conversion of about 60–70% was evaluated through the gravimetric method. Straight after the reaction, 2 g of solution was precipitated in 20 mL MeOH and the solid was recovered and completely dried at 100 °C under vacuum for 48 h. Solid powder was recovered from the solution through several cycles of extraction washing with THF for 10 min and centrifuging at 6000 rpm for 10 min in order to remove the PMMA homopolymer and the unreacted MMA. The powder was dried at 50 °C in a vacuum oven for 48 h and finely crushed by using an agate pestle-mortar set. The final product was a PMMA-g-HAP hybrid nanocomposite, labeled as gHAP. The first collection of solid layers was called PMMA/vHAP nanocomposite. It was noticed that the PMMA homopolymer tended to dissolve in the solvent phase; therefore, it was partially removed in the first centrifugation.

To confirm the reproducibility of the synthetic methods and to obtain enough PMMA-g-HAP for the whole experiment, several batches of the synthetic procedure of PMMA/vHAP were conducted.

#### 2.2.3. Preparation of PMMA Bone Cements Modified with HAPs

The powdered part of the bone cements was prepared according to two steps. In the first step, unmodified PMMA bone cement powder (Table 1: powder0) was prepared, with a total weight of about 120 g. It comprised fine PMMA powder (94.5%), nano ZrO_2_ (5%) and BPO (0.5%). It was mixed with a mini-electronic blade mixer for 30 s, as per the composition in Table 1. In the second step, 12 g of unmodified powder was mixed with every kind of HAP (oHAP, vHAP, gHAP) at each loading of 5, 10 and 15% of in an agate mortar for several mins. Finally, the mixtures of bone cement powder modified with HAP were stored in a PE bag at 8 °C until use.

The components of the liquid comprised MMA (49 g), DMPT (1 g) and 90 ppm of MeHQ. A 2000 ppm MeHQ stock solution (20 g of MMA, 39.5 mg of MeHQ, 200 mg of DMPT) was prepared in order to adjust the concentration of MeHQ to 90 ppm in the liquid part. Both the powder and liquid parts of the bone cements were stored in a vial at 8 °C until use.

In this study, the powder-to-liquid weight ratio was kept the same, at 2:1. Prior to mixing, two parts of bone cements were conditioned at 23 °C in a room for at least 2 h. The weights in grams of the bone cement powder/liquid were 9/4.5 and 3/1.5 for the rectangular-shape and cylindrical-shape samples, respectively. A stopwatch was started when the liquid was poured completely into the bowl containing the powder part; an inox spatula was used to mix the two parts together at room temperature for about 90 s. The obtained cement paste was inserted into a rectangular Teflon mold with a thickness of 3.2 mm, which was sandwiched between two thin Teflon sheets and two outer stainless steel plates for 50–60 min under a clamping pressure of about 4 MPa. For the cylindrical shape, the cement paste was inserted into a plastic syringe with dimension of 10 mm in diameter and filled to 50 mm in length under about 4 MPa pressure for 2 min. All the prepared samples were kept at room temperature for at least 7 days before characterization.

### 2.3. Characterization

#### 2.3.1. Setting Properties of Bone Cements

The setting properties of the HAP-modified bone cements were measured based on ISO 5833:2002. The cement pastes with different kinds and contents of HAP were prepared as described in Section 2.2.3. Each cement paste was inserted into a plastic syringe with diameter of 10 mm filled up to 50 mm in length under a pressure of 4 MPa for about 2 min; a thermocouple of Haake Polylab OS Instrument (Thermo Fisher Scientific, Waltham, MA, USA) was inserted at least 4 mm into the cement paste through the tip of the syringe. Recording of the temperature and time (using the Haake Polylab OS and software suite, Thermo Fisher Scientific, Waltham, MA, USA) were started when the liquid completely poured into the powder. The temperature of the bone cement inside was actually recorded once the paste had already been in contact with the thermocouple tip, otherwise the recorded temperature was ambient temperature (T_amb_). The maximum and setting temperatures (T_max_, and T_set_) were measured and calculated as described in the ISO 5833:2002 standard. The dough time (t_dough_) can be determined as the time when the cement mixture no longer adheres to rubber gloved fingers. The maximum time and the setting time (t_atmax_, t_set_) can be evaluated as the times when the maximum and setting temperatures were reached the working phase or handling time is the time interval from t_dough_ to t_set_, meaning that the cement paste in the dough state could be handled manually in this interval [16,30].

#### 2.3.2. Other Characterizations

The tensile properties of the bone cement samples were conducted on a universal mechanical testing machine (Zwick 2.5kN, Ulm, Germany), according to the ISO 527-1:2012 standard, tensile test for rigid plastic composites, at a crosshead speed of 5 mm/min. The specimens were prepared from cured bone cement sheets by laser cutting into rectangular shapes (beams) with dimensions of 75 × 5 × 3.2 mm^3^. The bending properties of the bone cements were analyzed in a Zwick 2.5 machine, according to the ISO 5833:2002 standard (annex E), with specimen dimensions of 75 × 10 × 3.2 mm^3^ and across-head speed of 5 mm/min. The compressive properties of the bone cements were conducted on a universal testing machine (Instron 5582-100kN, Norwood, MA, USA) at a cross-head speed of 1 mm/min, cylindrical specimens with 10 mm in diameter and 12 mm height, according to ISO 5833:2002. Each mechanical property was averaged from a set of three or five specimens for a bone cement sample.

To investigate the effect of wet conditions on the bending properties, some selected HAP-modified bone cements in a rectangular beam were soaked in standard saline solution at 25 ± 3 °C for 72 h. After soaking, the specimens were gently cleaned with towel paper and naturally exposed in open air for 2–3 h before the bending test.

FTIR spectra of all the samples were performed on a Fourier transform infrared spectrometer (Nicolet/Nexus 670, Madison, WI, USA) with 32 scans, a resolution of 4 cm^−1^, and in wavenumbers ranging from 400 to 4000 cm^−1^ at room temperature. The XRD patterns of the HAPs were examined on a Siemens D5000 X-ray diffractometer (Siemens, Munich, Germany) with 2θ in the range of 2–80° and CuKα radiation (45 kV, 40 mA, λ  =  1.5407 Å). The morphology of the bone cements was observed using a Hitachi field emission scanning electron microscope (FESEM, S-4800, Hitachi, Tokyo, Japan) at an electron acceleration voltage of 5 kV. The specimens were coated with Pt.

Thermal Gravimetric Analysis (TGA) of the HAPs and bone cements were carried out using a NETZSCH TG 209F1 Libra instrument (Netzsch, Munich, Germany) to study the thermal degradation properties under nitrogen gas. with a flow rate of 40 mL/min, from room temperature to 700 °C, with a heating rate of 10 °C/min and a specimen weight of about 10 mg. A differential scanning calorimeter (DSC) of the bone cement samples was conducted on a NETZSCH DSC204F1 (Netzsch, Munich, Germany) instrument under a nitrogen gas at flow rate of 40 mL/min, from 30 to 160 °C, with a heating rate of 10 °C/min and a specimen weight of about 10 mg. All the bone cement samples were kept at 160 °C for 30 min and then cooled to room temperature for clearing their thermal history.

### 2.4. Statistical Analysis

The data analysis was performed by using Microsoft Excel 2016 (Microsoft, Redmond, WA, USA). Where appropriate, the results are expressed as mean ± standard deviation (SD, “n” method). The T-test function with parameters of “one-tail” and “paired-sample for means” was used to analyze significant differences between a pair of bone cement samples. A significant difference was verified if the *p*-values < 0.05.

## 3. Results and Discussion

### 3.1. Characterization of HAPs

#### 3.1.1. FTIR Spectra of HAPs

Figure 1 represents the FTIR spectra of the three kinds of HAPs and the pristine VTMS liquid. Figure 1a shows the characteristic absorption bands of the original hydroxyapatite (oHAP), the stretching absorption bands (ν) of the PO_4_^3−^ at 1094, and 1036 cm^−1^ (denoted as ν_1_,_2_), the bending absorption (δ) at 604 and 567 cm^−1^ (denoted as δ_1_,_2_) and the stretching vibration (ν) bands of the carbonate residual at 1455 and 1420 cm^−1^ [29,31]. In the spectrum of the vHAP (Figure 1b), apart from the absorption bands of the hydroxyapatite, the specific absorption bands of the hydrolyzed VTMS can be observed when compared with the spectra of the oHAP and the pristine VTMS (Figure 1d). It should be noted that the residual VTMS and PMMA homopolymer were removed from the vHAP and gHAP through washing and extraction processes, respectively. Therefore, the appearances of the ν(CH) and ν(CH_2_) groups at 2928 and 2848 cm^−1^, respectively, the ν(C=C) at 1607 and the ν(Si–C) at 762 cm^−1^ (Figure 1b) indicate that the hydrolyzed VTMS silane was successfully grafted onto the surfaces of the HAP particles as per the scheme of proposed reactions in Figure 2a,b [32]. Figure 1c is the spectrum of the gHAP, which clearly shows the specific absorption bands of the PMMA grafted onto surfaces of the HAP particles, such as δ(C=O) at 1729 and 754 cm^−1^, δ(CH_2_) at 1452 cm^−1^, ν(–CH_3_, OCH_3_ and CH_2_) vibrations at 2997, 2953 and 2854 cm^−1^, δ(CH_3_) at 1453 cm^−1^ and ester C–O–C bands at 1243 and 1143 cm^−1^ [33,34]. The PMMA grafting reaction is displayed in Figure 2c. The band of the ν(C=O) in the spectra of the gHAP is within the typical wavenumber position of the methyl ester carbonyls (not bound) of the PMMA chains. Furthermore, the bands ν(PO_4_^3−^) in the spectra of the gHAP are the same as those in the spectra of the oHAP. These results may have been due to the fact that the grafted PMMA content and its molecular weight in gHAP were all rather low; therefore, the improvement in the PMMA graft content and molecular weight should be further investigated.

#### 3.1.2. TGA of HAPs

Figure 3 displays the TGA and derivative TG (DTG) curves of the oHAP and vHAP samples in the temperature range from 30 °C to 700 °C. Chemically, HAP is a thermal stable compound; however, the total weight loss of the oHAP was 4.77% due to the desorption of moisture and decarbonation [29]. The TGA and DTG curves of the vHAP presented a slight difference with those of the oHAP, especially from 400 °C to 700 °C because of the decomposition of the grafted VTMS on the surfaces of the HAP particles. In other words, this qualitatively demonstrates that the hydrolyzed VTMS was grafted onto the surfaces of the HAP nanoparticles (as shown in the reactions in Equation (1)). The organic grafting content on the surfaces of the HAPs can be evaluated from the TGA measurement data. When Liu Z. et al. [35] treated HAP with 3-trimethoxysilyl propyl methacrylate (MPS), using TGA measurement, the weight loss from 400–600 °C in the TGA curve of the MPS-treated HAP was assigned for MPS grafting content. Kang T. et al. [36] also assigned the weight loss from 350–700 °C in the TGA curve of treated HAP for the organic graft content on the surfaces of treated HAP particles. In our investigation, VTMS has lower molecular weight than MPS, and a low loading of VTMS was used (1 mmol of VTMS per 1 g of HAP). In the range of 400–700 °C, the degradation of oHAP was still observed. Furthermore, the proposed degradation reaction in Figure 2d shows that vHAP degrades into pure HAP, SiO_2_ and gases derived from VTMS (with a molecular formula equivalent to OC_2_H_6_, 46.07 mol/g). It can be proposed that the difference between the weight losses from 400 °C to 700 °C of the vHAP and oHAP can be assigned to the degraded organic part (OC_2_H_6_) [37]. As shown in Figure 3, this value is calculated as 1.48% − 0.82% = 0.66%. Theoretically, the molar amount of VTMS grafting on the surfaces of the HAPs can be calculated as Equation (1) with a value of 0.143 mmol/g (of HAP) or a percentage of VTMS grafting of 1.52 wt.%, as in Equation (2), where 106.15 is molecular mass of (OH)_3_SiCH = CH_2_ [38]. In comparison with the VTMS loading of 1.0 mmol/g, the VTMS grafting content of 0.143 mmol/g was quite low. This means that the loading dose of the VTMS should be further increased.
(1)$$\text{Molar amount of VTMS grafting}=\frac{0.66\%}{46.07}\;\times 1000\;=0.143\;\text{(mmol/g)}$$
Percentage of VTMS grafting (wt.%) = 0.143/1000 × 106.15 = 1.52 wt.%(2)

For the PMMA/vHAP nanocomposite (collected after the first centrifugation) and the gHAP sample, the TGA and DTG curves (in Figure 4) exhibited two stages of decomposition. The first decomposition stage occurred from 30 °C to 325 °C, due to the evaporation of moisture (similar to oHAP and vHAP), residual monomer and PMMA oligomers. The second stage was the decomposition of the organic moieties (PMMA and VTMS), occurring from 325 °C to 750 °C, which can be observed more clearly in the DTG diagrams of the PMMA/vHAP nanocomposite and the gHAP in Figure 4b. The weight loss in this stage was 12.92% for the PMMA/vHAP, which can be attributed to the total PMMA content in the nanocomposite. The weight loss of 5.85% in the second stage for the gHAP sample (from 325 °C to 750 °C) can be attributed to the grafted PMMA content and the strong physical attachment of the PMMA molecules on the surface of the gHAP nanoparticles [35,36]. In this case, the weight losses due to the degradation of the VTMS and the HAP were mostly neglected.

#### 3.1.3. Morphology and XRD Spectra of HAPs

Figure 5 represents the FESEM images and XRD patterns of the three kinds of HAPs. The FESEM images (Figure 5a–c) demonstrate the rod-like shaped of the HAP nanocrystals, with an average length and width of about 80 nm and 20 nm, respectively. Figure 5d,e are the XRD patterns of the oHAP and vHAP samples, which indicate the same diffraction peaks at 2θ angles. Based on the XRD analysis, these patterns are fitted well with JCPDS 004-09-0432 standard for hexagonal HAP with lattice constants as a = b = 9.418 Å; c = 6.884 Å and α = β = 90°; γ = 120° and lattice faces as follows: 10.88° (100), 25.9° (002), 28.94° (210), 31.86° (211), 32.96° (003) 34.1° (202), 39.91° (130), 46.73° (222), 49.52° (213) and 53.24° (004) [31,39]. This means that the crystalline structure of the HAP nanocrystals does not change after surface treatment with VTMS [38]. Figure 5c,f are the FESEM image and XRD pattern of gHAP, respectively. As can be seen, the surfaces of the HAP nanocrystals become smooth. This change in the morphology of the gHAP sample probably resulted from the grafting of the PMMA through the surface-initiated reaction of MMA. The XRD pattern of the gHAP also displayed a broad peak in 2θ region from 5 to 20°, which may have been due to the amorphous phase of the PMMA chains grafted onto the HAP particles.

### 3.2. Properties of PMMA Bone Cements Modified with HAPs

#### 3.2.1. Setting Properties

It is well known that benzoyl peroxide (BPO) is commonly used as an initiator and that N,N dimethyl *p*-toluidine (DMPT) is used as a co-initiator for bone cement. DMPT can interact with BPO at a low (room) temperature to generate free radicals, at which point the curing of bone cement starts. In bone cement components, the BPO initiator is in the powdered part and the DMPT is in the liquid part. When mixing the two parts, a curing reaction (solidification) occurs to form a bone cement in solid form after a certain time [24,40]. In this study, the powder parts were mixtures of PMMA beads, ZrO_2_, BPO without and with three kinds of HAPs (oHAP, vHAP, gHAP); the liquid part was the same for all the samples and the powder/liquid weight ratio was 2:1. Figure 6 plots the temperature versus time during the solidification of the PMMA bone cements modified with three kinds of HAPs at three loading levels (5, 10, 15 wt.% compared with 0 wt.% of HAP for unmodified powder weight). From the recorded data and the plotting curves, the setting properties of the HAP-modified bone cements (HAP-BCs) were determined and are represented in Table 2. Figure 5 clearly indicates that increasing the HAPs loading reduced the maximum temperature (T_max_) and remarkably prolonged the setting time (t_set_) of the HAP-BCs while curing. Arithmetically, Table 2 shows that the T_max_ values of oHAP-BC series slightly decreased through 64.9, 64.0, 62.2 and 61.5° and the t_work_ values significantly increased through 4.3, 6.1, 8.1, 9.5 min when increasing the oHAP loadings from 0 to 5, 10 and 15 wt.%, respectively. The vHAP-BCs and gHAP-BC bone cement groups also exhibited the same tendency. Based on the bone cement formulae, this can be simply explained by the fact that increasing HAPs loading leads to the lowering (dilution) of the BPO initiator concentration, thus prolonging the induction time of the initiation of polymerization (when the concentration of the generated free radicals is high enough above that of the MeHQ inhibitor concentration). The lower concentration of BPO also resulted in a slightly lower T_max_ of the HAP-BCs than that of the unmodified bone cement (without HAP). This was due to slower rate of exothermic polymerization and heat loss due to the prolonged time. The longer working/handling time and the lower T_max_ values of the HAP-BCs are very significant for handling with bone cement in longer intervals, as well as reducing thermal necrosis in clinical application [41,42,43].

#### 3.2.2. Morphology

Figure 7 displays the FESEM images of the fractured cross-section of the unmodified-BC (containing 5% of ZrO_2_ and 0 wt.% of HAP) and bone cements modified with 10 wt.% of oHAP, vHAP and gHAP nanoparticles, labeled as oHAP10-BC, vHAP10-BC and gHAP10-BC, respectively, at some magnifications. In general, the morphology of the above bone cement samples presented four distinct material phases: prepolymer PMMA beads, polymerized PMMA and radiopacifier/filler and pore phases [1]. The radiopacifier/filler particles dispersed into the polymerized phase, which appeared as white specks at low magnification. At a higher magnification, nanoparticles were easily observed; the ZrO_2_ nanocrystals were nearly round in shape (Figure 7c, ×50 k) and the HAPs nanocrystals were rod-like in shape (Figure 7e,f,h, ×50 k). When mixing HAP-modified bone cements, the ZrO_2_ and HAPs particles first dispersed in the liquid phase and then embedded into polymerized phase. Figure 7a,d,g,j (left) show that the prepolymer PMMA beads appeared in dark grey areas without the white specks. The FEFEM images of the HAP-modified bone cements (×150) also show that the prepolymer bead sizes ranged from 30 to 110 µm. Figure 7c indicates that the white specks are the clusters of nano ZrO_2_ nanoparticles agglomerated from several to tens of primary particles. For the oHAP-BC samples, Figure 7f indicates that the radiopacifier/filler particles appeared in larger clusters, each of which comprised several tens of primary nanoparticles of ZrO_2_ and HAPs.

It can also be observed some micron gaps between the oHAP clusters and the PMMA matrix (the gap is viewed as polymer-filler interphase thickness) [44]. Because of the difference in affinity, the oHAP particles were unable to bond with the PMMA matrix, corresponding to a weak filler-polymer adhesion and large interphase thickness. By contrast, Figure 7i,l show that the vHAP and gHAP particles seemed to be in good contact with the polymerized matrix; smaller interphase thickness and clusters were observed. These demonstrate that the vHAP and gHAP particles were better dispersed and adhered with the matrix, corresponding to their better affinity and stronger filler-polymer interactions, compared to the oHAP. As described in Section 3.1, the TG and FTIR analyses demonstrated the presence of vinyl groups on the surfaces of the vHAP particles, which could be copolymerized with MMA to form a PMMA-g-HAP hybrid, as well as the presence of grafted PMMA molecules on the surfaces of the gHAP particles (reaction scheme in Figure 2c). Therefore, the higher wettability and organo-affinity of both the vHAP and gHAP particles can improve their dispersion in MMA liquid while mixing and in the polymerized matrix after curing [45,46].

Figure 8 represents the FESEM images of a fractured cross-section of the gHAP-BC group with gHAP loadings of 5, 10 and 15 wt.% (gHAP5-BC, gHAP10-BC and gHAP15-BC). Figure 8 indicates that clusters of gHAP and ZrO_2_ nanoparticles tended to be larger in size, even though the HAP was grafted with PMMA to become more organically affinitive. Nevertheless, the higher-magnification FESEM images demonstrate that functionalized HAP was dispersed better in the polymerized matrix than the original HAP particles.

#### 3.2.3. FTIR

Figure 9 represents the attenuated total reflectance (ATR) FTIR spectra in absorbance mode in the region 2000–500 cm^−1^ of the bone cement samples modified with gHAP loadings of 0, 5, 10 and 15 wt.%, labeled as unmodified-BC, gHAP5-BC, gHAP10-BC and gHAP15-BC, respectively. It should be noted that all the samples contained the same loading, 5 wt.% of ZrO_2_. In the FTIR analysis, the highest peak (at 1145 cm^−1^) is normalized with baseline to 1.000. The FTIR spectrum of the unmodified-BC showed a broad absorption, in the region 750–500 cm^−1^, centered with the peak at 571 cm^−1^ due to the broad and strong absorption of ZrO_2_ in this range [47,48]. Other peaks were assigned for specific groups of the PMMA, such as the stretching (ν) of the C=O group at 1725 cm^−1^, the bending vibrations (δ) of the CH_3_ and CH_2_ groups at 1483, 1435 and 1386 cm^−1^ and the ν(C–O) at 1240 cm^−1^, 1145 cm^−1^ [49]. The weak peak at 1638 cm^−1^ can be imputed to δ(OH) and ν(C=C); however, in the case of the cured bone cements, it can be assigned to ν(C=C) due to an MMA residual. In general, these specific groups of PMMA appearing in the spectra of the gHAP-BC samples seem unchanged when adding gHAP into PMMA bone cement. By contrast, Figure 6 evidently shows that the band of ν_2_(PO_4_^3−^) at 1034 cm^−1^ appeared in the FTIR spectra of the gHAP-BC samples with an absorbance height (Ht) that increased along with the increase in the HAP loading, while another ν_1_(PO_4_^3−^) peak at 1093 or 1094 cm^−1^ appeared only if the HAP loading was higher than 5 wt.%, due to its weak absorption and overlapping. The double peaks of δ_1,2_(PO_4_^3−^) also appeared in the FTIR spectra of the gHAP-BC samples: the first was around 600 cm^−1^ and the second was around 561–567 cm^−1^. The FTIR spectrum of the oHAP (Figure 1a above) displayed peaks of ν_2_(PO_4_^3−^) and δ_1,2_(PO_4_^3−^) at 1036, 604 and 567 cm^−1^, respectively. In comparison, the specific vibration bands of the PO_4_ groups appeared in the spectra of the gHAP-BC samples as shifted to lower wavenumbers: the ν_2_(PO_4_^3−^) peak was shifted to about 2–7 cm^−1^, the δ_1_(PO_4_^3−^) peak was shifted by about 4–5 cm^−1^ and the δ_1_(PO_4_^3−^) peak was shifted by about 1–6 cm^−1^. The superimposition between ν_2_(PO_4_^3−^) and ν(C–C) at 1063 cm^−1^ of the back-bone of the PMMA also led to the formation of a new peak or shoulder at 1057–1060 cm^−1^ in the spectra of the gHAP-BCs, with a shift of about 3–6 cm^−1^.

Figure 10 presents the FTIR spectra of the bone cements modified with 15 wt.% of oHAP, vHAP and gHAP nanoparticles, labeled as oHAP15-BC, vHAP15-BC and gHAP15-BC, respectively. Figure 9 also shows the shift of about 5–7 cm^−1^ compared with that of the oHAP (Figure 1a). These shifts were attributed to the strong polar interaction between the PO_4_ groups of the gHAP nanoparticles and the ester groups of the polymerized matrix [36]. The higher organic affinity of the gHAP and vHAP nanoparticles made them disperse into the polymerized matrix with smaller interface thickness, leading to stronger filler-polymer interactions or higher shifts in the ν_2_(PO_4_^3−^) and δ(PO_4_^3−^) peaks when compared with the FTIR spectrum the oHAP nanoparticles.

#### 3.2.4. Mechanical Properties

Figure 11 plots the column chart of the bending strength (BS) and bending modulus (BM) of the HAP-BCs at different loadings of HAP, showing that the bending strength (BS) of the three groups of HAP-BCs decreased as a monotonical function of HAP loading (0 wt.% for unmodified bone cement). The BM of the oHAP-BC group (the pink columns) decreased from 2.31 to 2.06 GPa along with the increase in the oHAP loading from 0 to 15 wt.%; however, the BM of the vHAP5-BC and gHAP5-BC groups increased slightly to maximum values (2.41 and 2.38 GPa) at 5 wt.% of HAP loading and then tended to decrease. The important result is that the bending properties of the bone cements using functionalized HAPs (vHAP-BC, gHAP-BC) were significantly higher (*p*-value < 0.05) than those of the bone cements using the oHAP (oHAP-BC).

Figure 12 displays the elongations at break (EB), tensile strength (TS) and tensile modulus (TM) of three series of HAP-BCs as a function of the HAP loadings. Figure 12a shows that the elongations at break (EB) were in the range from 2.4 to 5.2%, indicating the brittle behavior of the PMMA-based bone cements. Regardless of the type pf HAP used, the EB and TS of the HAP-modified bone cements monotonically decreased from 0.97 to 0.82 MPa as the HAP loading increased from 0 to 15 wt.%. The TMs of the vHAP5-BC and gHAP5-BC reached their maximum values (1.01 and 1.09 GPa) at 5 wt.% of HAP loading and then tended to decrease. Similar to the bending properties, the tensile properties of the vHAP-BC or gHAP-BC groups were higher than those of the oHAP groups in comparison with the same HAP loading.

As reported in previous studies, the enhancement of the mechanical properties of polymer nanocomposites can usually be achieved at very low filler loadings, e.g., in the range of 1–5 wt.% [25,26], or somewhat higher, only in the case of a strong interaction within the filler-polymer matrix. In this study, the lowest filler loading was 10 wt.% (included HAP and ZrO_2_). Moreover, the fillers (HAP and ZrO_2_) dispersed into the polymerized PMMA phase (from MMA); thus, the real percentages of fillers in this phase were as high as 16.0, 21.4 and 25.8 wt.%, corresponding to the samples containing 5, 10 and 15 wt.% of HAP, respectively. Therefore, increasing the loading of the HAP nanoparticles led to an increase in the size and number of the agglomerated clusters generated, which act as crack growth centers, and fracture occurred at a critical deformation, lowering the bending and tensile strengths of the bone cements [50]. However, in the case of using organically functionalized HAP nanoparticles (vHAP and gHAP), the higher mechanical properties of the vHAP-BC and gHAP-BC groups compared with the oHAP group were obtained. As mentioned in Section 3.1, the presence of vinyl groups and grafted PMMA molecules on the surfaces of the vHAP and gHAP nanoparticles, respectively, promoted the higher organic affinity of the vHAP and gHAP nanoparticles, which improved their dispersion and interaction in the polymerized matrix better than the original HAP nanoparticles (oHAP), thus improving the tensile and bending strengths. When using a low loading of functionalized HAP nanoparticles (5 wt.%), homogeneous dispersion can be achieved, which can improve the moduli of bone cements [51].

Figure 13 evidently shows that the compressive strength (CS) and compressive modulus (CM) of the HAP-modified bone cements (modified with 3 kinds of HAP) significantly improved in the presence of the HAPs. It was noticeable again that the compressive properties of the vHAP-BC and gHAP-BC groups were higher than those of the oHAP-BC group at each HAP loading. The results were different in the tensile and bending tests; the compression force shortens the interphase thickness, leading to a strengthening of the nanoparticles/polymer interaction. Therefore, HAP nanoparticles as well as functionalized HAP nanoparticles can reinforce the compressive properties of bone cements [52]. The obtained results are in agreement with the findings of M. E. Islas-Blancas et al. [53].

According to the ISO 5833 standard, the requirements of the BS, BM and CS of bone cement are higher than 50 MPa, 1.8 GPa, and 70 MPa, respectively, which are plotted as the black lines at 50 MPa of BS and at 1.8 GPa of BM in Figure 10 and at 70 MPa of CS in Figure 12a. According to that standard, the oHAP-BC group included only one sample (oHAP5-BC that contained a low loading of oHAP, at 5 wt.%) that met the requirements, while all six samples of the vHAP-BC and gHAP-BC groups met the ISO 5833 requirements for bending and compressive properties, indicating the effectiveness of organic functional HAPs when applied in bone cements. Although there is no requirement for TS, TM and CM in the ISO 5833 standard, the obtained results for the tensile properties and compressive modulus in this study are in the tolerance range observed for commercial bone cements in other works [25,40,54,55,56,57].

Table 3 represents the variations in the BM and BS of unmodified and oHAP10-, vHAP10-, and gHAP10-modified bone cements before and after soaking in saline solution at 25 ± 3 °C for 72 h. Previous studies showed the high initial water uptake (1–2%) of acrylic bone cements and HAP/PMMA bone cement during the first few days of soaking [4,5,58]. With 72 h of soaking, the water absorption into the bone cement samples was remarkable. Table 3 shows that the BS of the oHAP10-BC sample significantly reduced (7.4%, * *p* < 0.05, and its BM only slightly reduced (1.4%, *p* > 0.10) after soaking. Meanwhile, the BS and BM of the other samples (unmodified-BC, vHAP10-BC, gHAP10-BC) only demonstrated a slightly declining trend after soaking. This decline trend could have been due to the absorption of water under wet conditions. The FESEM images in Figure 7 show that the vHAP and gHAP particles were better dispersed and adhered with the matrix than the oHAP particles, which may have reduced the porosity and water absorption ability of the bone cements. Therefore, the bending properties of the vHAP10-BC and gHAP10-BC samples were less reduced. In fact, the high *p* values (from 0.14 to 0.18 > 0.05) suggest that the bending properties of the vHAP10-BC and gHAP10-BC bone cements were not significantly varied. According to the requirements of ISO 5833, the BS of the oHAP10-BC sample was lower than the minimum value required in (50 MPa), but those of the vHAP10-BC and gHAP10-BC were higher. This indicates the prospective application of gHAP- and vHAP-modified bone cements in wet environments.

#### 3.2.5. TGA and DSC

Figure 14 represents the TGA diagrams and offset DTG curves of the PMMA beads (Mw of 120 kDa) and PMMA bone cements modified with 10 wt.% oHAP (labeled as oHAP10-BC) and with 5, 10 and 15 wt.% gHAP (labeled as gHAP5-BC, gHAP10-BC and gHAP15-BC, respectively). Figure 14 generally indicates that the thermal degradation of the unmodified BC and HAP-BC samples was similar to that of the PMMA beads; all underwent one minor stage and two major stages of thermal degradation. The minor stage (I) mainly occurred from 140 °C to 225 °C with low weight loss, which was attributed to the monomer residual, absorbed water and weak polymer segments with head-to-head linkages. The two major stages (II) and (III) were mainly in the regions of 225–325 °C and 235–450 °C, with higher weight losses due to the degradation of the unsaturated vinyl ends and the main chain scission of the PMMA macromolecules, respectively [59,60,61]. Figure 14 evidently shows that the TGA curves of the unmodified BC and oHAP10-BC shifted slightly to a higher temperature, meaning that the nano ZrO_2_ nanoparticles and the mixture of ZrO_2_/HAP did not significantly improve the thermal stability of the PMMA.

Figure 15 demonstrates that the incorporation of the gHAPs only changed the residual weights of the gHAP-BCs (at 700 °C), which increased as the gHAP loadings increased [62]. It is well known that HAP and ZrO_2_ are thermally stable (their weight losses up to 700 °C are lower than 10%), their concentration in bone cement systems is about 10 wt.%), thus their contribution to the weight loss of the bone cement system can be approximately neglected (10% × 10% = 1%). In other words, the inorganic phase is considered an inert component in thermal degradation; the physical interaction is not strong enough for the improvement of thermal stability in polymer matrices. The degradation of the HAP-modified PMMA bone cements was mainly related to the PMMA matrices, including the PMMA beads (prepolymer) and the polymerized PMMA.

Table 4 represents the results of the determination of the weight loss in three stages (WL_1_, WL_2_ and WL_3_) and the final residual weight (W_f_ at 700 °C) from the TGA curves of investigated samples. It can be seen that the WL_1_ values of the bone cements (average of about 5.0%) were higher than those of the PMMA beads (3.94%). The lower values of the WL_1_ of the PMMA were due to the partial removal of the residual MMA content from the pulverizing and drying processes. However, the WL_1_ of the gHAP-BC series slightly decreased as the HAP loading content increased. In other words, the incorporation of gHAP does not significantly reduce the residual MMA content in cured bone cements.

Figure 16a represents the DSC diagrams of the PMMA bone cement samples, labeled as unmodified-BC, gHAP5-BC, gHAP10-BC and gHAP15-BC. Figure 16b represents the DSC diagrams of the PMMA bone cements, labeled as oHAP10-BC, vHAP10-BC, gHAP10-BC. It should be noted that all the bone cement samples contained the same 5 wt.% of ZrO_2_ (ZrO_2_ was initially mixed in the powder parts) for investigating the effect of the HAPs; here, the effect of the ZrO_2_ was supposed to have been the same for the all samples. In comparison with the T_g_ of the PMMA beads (mid-point T_g_ of 105 °C for Mw of 120 kDa, provided by Sigma Aldrich), the glass temperatures of the HAP-modified bone cements were lower, with T_g_ values in the range of 94–99 °C. This phenomenon was probably related to the presence of residual MMA, a common issue when synthesizing acrylic bone cement that can hardly be avoided. Residual MMA acts as a plasticizer that increases the mobility of PMMA molecules and thus reduces the T_g_ of PMMA. However, the unique T_g_ indicates the homogeneity of the PMMA prepolymer and polymerized phases in the HAP-BCs samples. In comparison with the T_g_ (mid-point) of the unmodified BC (0 wt.% of gHAP), Figure 16a shows that the T_g_ of gHAP-BCs increases from 96.25 °C to 98.65 °C (increment of 2.4 °C) along with the increase in gHAP loading from 0 to 10 wt.% and then decreases to 95 °C when the gHAP reaches 15 wt.% (lower than the Tg of unmodified-BC). This increment indicates that the incorporation of gHAP contributes to a change in the original interaction strength in the polymer matrix and the mobility of the PMMA chains, which affect the behavior of the PMMA matrix when changing from the glassy (brittle) to the ductile stage. When the gHAP concentration is low enough (10 wt.%), the gHAP-BC exhibits better dispersion (as shown in Figure 7a,b), reinforcing the interactions between the gHAP nanoparticles and the PMMA matrix, thus hindering the motions of the PMMA chains, increasing the T_g_ of the gHAP-BC [45,62]. When the loading of the gHAP reaches 15 wt.%, the agglomeration of the HAP particles can be observed, weakening the filler-polymer interactions, therefore reducing the T_g_ of the gHAP-BC. Figure 16b shows that the T_g_ values of the vHAP10-BC (97.1 °C) and gHAP10-BC (98.7 °C) were higher than those of the oHAP10-BC (94.5 °C). There are two possible reasons for the T_g_ of vHAP10-BC being higher than that of the oHAP10-BC sample. The first is related to the higher organic affinity of the VTMS-grafted layer on the surfaced of the HAP nanoparticles and the second is the in situ formation of the PMMA-g-HAP during the curing process, due to the copolymerization of the vinyl groups on the surface of the vHAP and MMA monomers, initialized by the BPO and DMPT initiators. Meanwhile, the interaction between the oHAP particles and the PMMA matrix was weaker than with the vHAP. This means that the mobility of the PMMA chains in the vHAP10-BC system was lower, leading to its higher T_g_ compared with the oHAP10-BC.

## 4. Conclusions

In this study, three groups of HAP-modified bone cements were prepared by using three kinds of HAP nanoparticles: original HAP (oHAP), VTMS-treated HAP (vHAP) and PMMA-grafted HAP (gHAP). All the cement samples contained the same 5 wt.% of ZrO_2_ as the radiopacifier. The obtained results indicated that HAP-modified bone cements exhibited longer setting times (10–15 min) and lower maximum exothermic temperatures while curing (64–61 °C) when increasing the HAP loading from 5 to 15 wt.%. The FTIR of the bone cement samples showed a polar interaction between the PO_4_ groups of the HAPs and the ester groups of the polymerized PMMA in the modified bone cements. The FESEM images indicated the better dispersion of the functionalized HAPs in the polymerized matrix than that of the oHAP. According to the ISO 5833 standard, the two samples in the oHAP-BC group that were modified with 10 and 15 wt.% of oHAP did not meet the ISO 5833 requirements for mechanical properties, while all six samples of the vHAP-BC and gHAP-BC groups (modified with 5, 10 and 15 wt.% of HAPs) did. This demonstrated the effectiveness of organic functional HAPs when applied in acrylic bone cements. The TGA measurement indicated that the thermal degradation of the HAP-modified bone cements was similar to that of the unmodified bone cement. The DSC measurement revealed the unique T_g_ of the gHAP-BCs. Their T_g_ values increased from 96.25 °C to 98.65 °C (∆ = 2.4 °C) when the gHAP loading increased from 0 to 10 wt.% and then decreased to 95 °C when the gHAP reaches 15 wt.%. The T_g_ values of the vHAP10-BC (97.1 °C) and gHAP10-BC (98.7 °C) were higher than those of the oHAP10-BC sample (94.5 °C). For prospective clinical applications, studies on other important properties, as well as in vitro and in vivo biological investigations of HAP-modified acrylic bone cements, should be performed in the future.

## Figures and Tables

**Figure 1 polymers-13-03860-f001:**
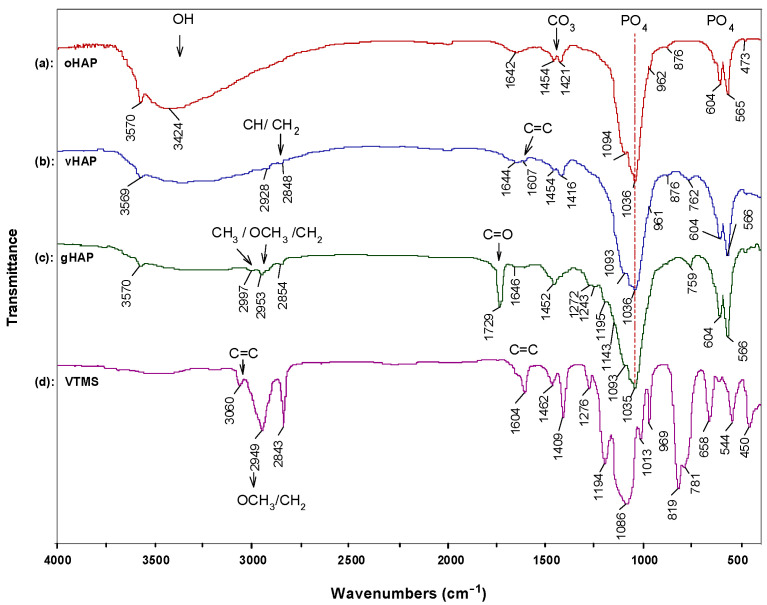
FTIR spectra of (**a**): oHAP, (**b**): vHAP, (**c**): gHAP and (**d**): VTMS.

**Figure 2 polymers-13-03860-f002:**
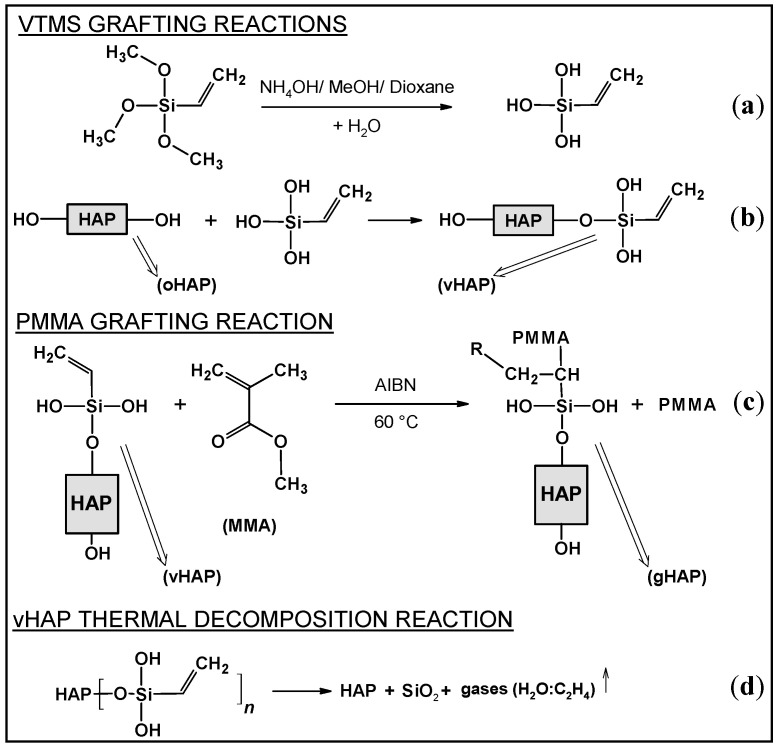
Schemes of (**a**): VTMS hydrolysis reaction, (**b**): VTMS grafting reaction onto the surface of HAP, (**c**): PMMA grafting reaction onto the surface of HAP, (**d**): Proposed thermal decomposition reaction of vHAP.

**Figure 3 polymers-13-03860-f003:**
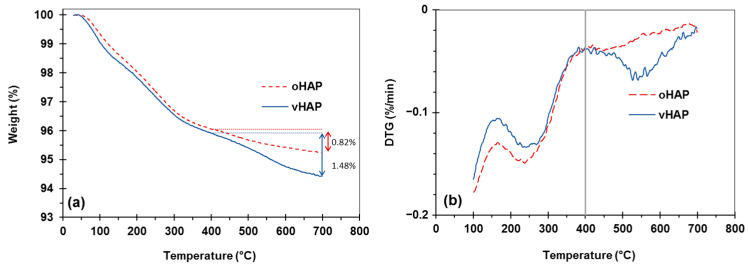
TGA and DTG curves of oHAP and vHAP; (**a**): TGA, (**b**) DTG.

**Figure 4 polymers-13-03860-f004:**
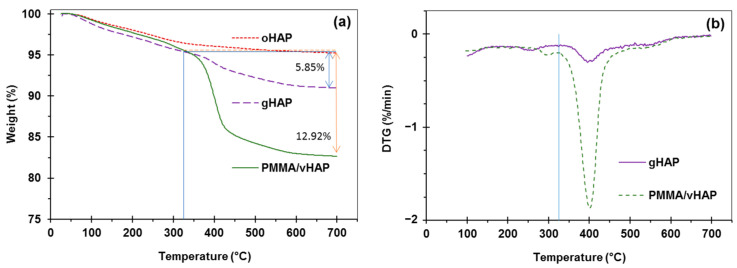
(**a**): TGA curves of PMMA/vHAP nanocomposite and gHAP (extracted from the nanocomposite), (**b**): DTG curves of PMMA/vHAP nanocomposite and gHAP. (Note: TGA curve of oHAP is used for comparison).

**Figure 5 polymers-13-03860-f005:**
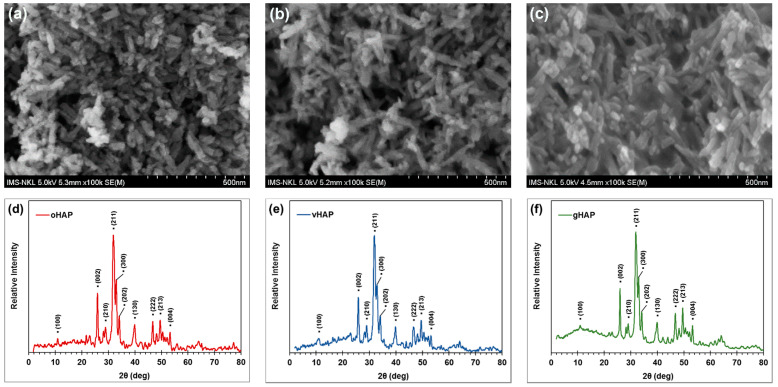
FESEM images of (**a**): oHAP, (**b**): vHAP, (**c**): gHAP; and XRD spectra of (**d**): oHAP, (**e**): vHAP and (**f**): gHAP.

**Figure 6 polymers-13-03860-f006:**
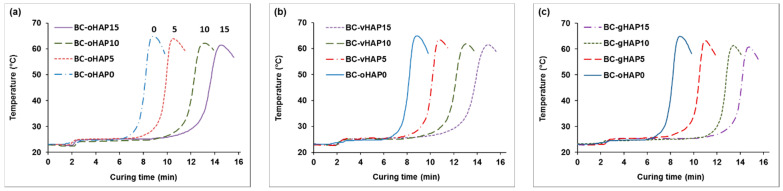
Curing temperature versus time of PMMA bone cements modified with 0, 5, 10, 15 wt.% of (**a**): oHAP, (**b**): vHAP, (**c**): gHAP.

**Figure 7 polymers-13-03860-f007:**
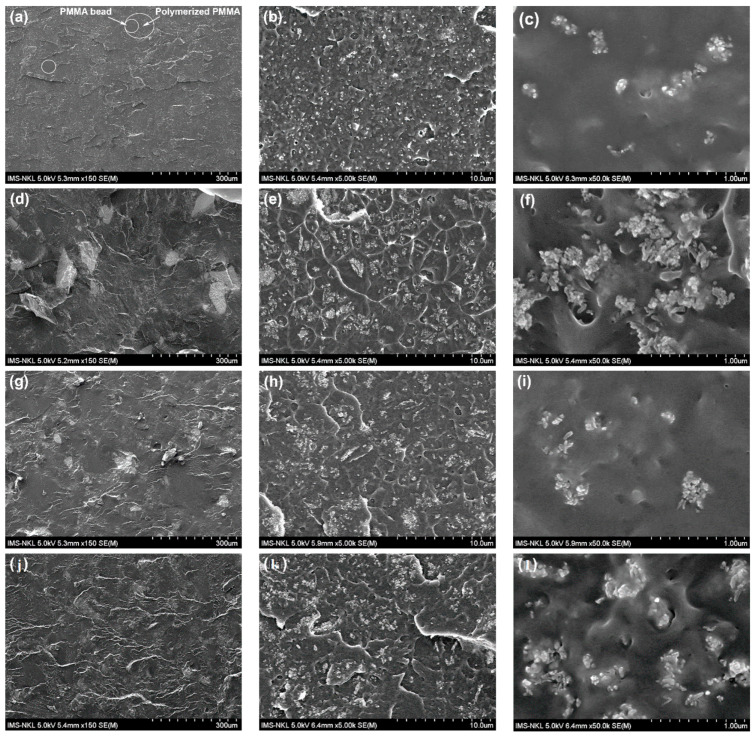
FESEM images of fractured cross-section of unmodified and HAP-modified bone cements: (**a**–**c**): unmodified-BC, (**d**–**f**): oHAP10-BC, (**g**–**i**): vHAP10-BC and (**j**–**l**): gHAP10-BC.

**Figure 8 polymers-13-03860-f008:**
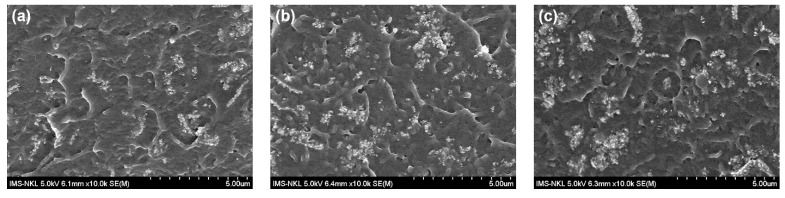
FESEM images of fractured cross-section of gHAP-BC group at different loadings of gHAP (**a**): 5 wt.%, (**b**): 10 wt.% and (**c**): 15 wt.%.

**Figure 9 polymers-13-03860-f009:**
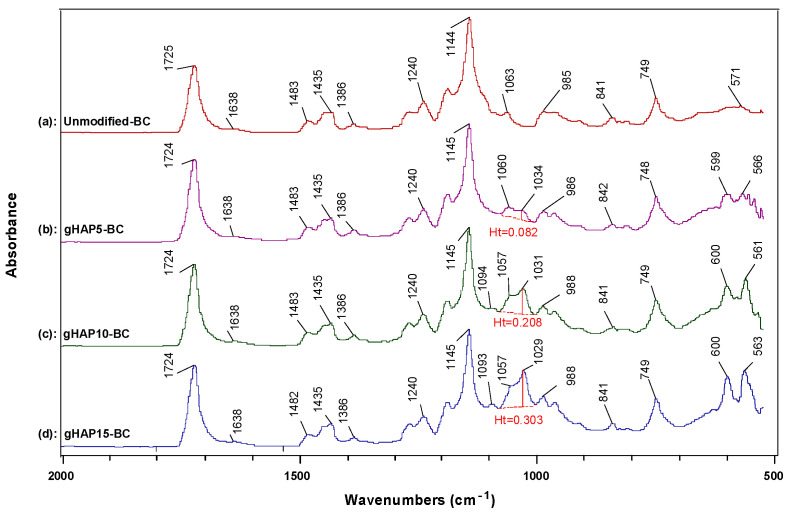
FTIR spectra of bone cement samples modified with 0, 5, 10, 15 wt.% loadings of gHAP, labeled as (**a**): unmodified-BC, (**b**): gHAP5-BC, (**c**): gHAP10-BC and (**d**): gHAP15-BC, respectively. (Note: all samples contained 5 wt.% of ZrO_2_).

**Figure 10 polymers-13-03860-f010:**
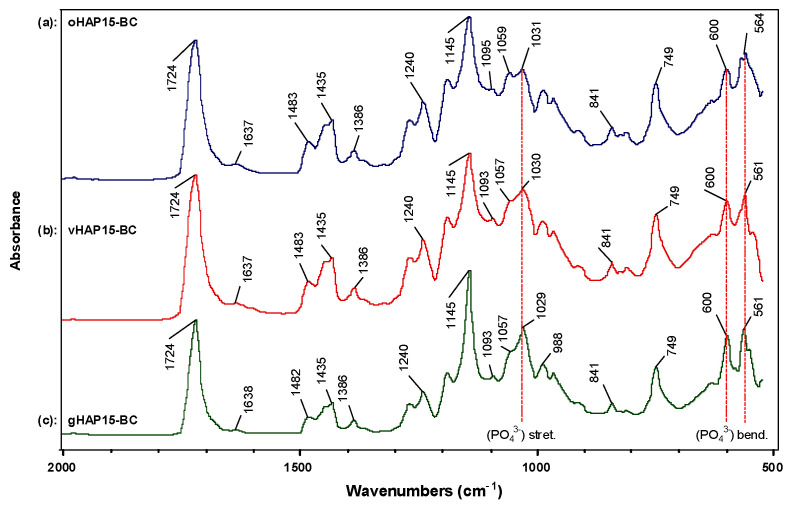
FTIR spectra of bone cements modified with 15 wt.% of oHAP, vHAP and gHAP, labeled as (**a**): oHAP15-BC, (**b**): vHAP15-BC and (**c**): gHAP15-BC, respectively.

**Figure 11 polymers-13-03860-f011:**
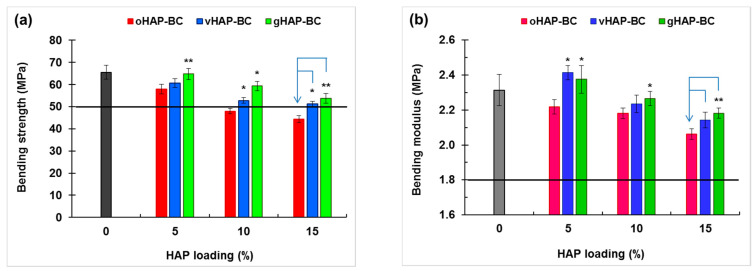
Bending properties of HAP-modified bone cements as function of HAP loadings. (**a**): bending strength and (**b**): bending modulus. (Result = mean ± SD, n = 5; one tailed *t*-test, * and ** indicates a significant difference and more difference from the oHAP-BC sample at the same HAP loading, * *p* < 0.05, ** *p* < 0.005).

**Figure 12 polymers-13-03860-f012:**
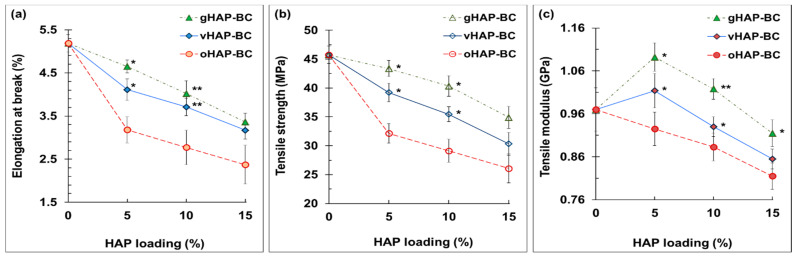
Tensile properties of HAP-BCs bone cements as function of HAP loadings. (**a**): elongation at break, (**b**): tensile modulus, (**c**): tensile strength. (Result = mean ± SD, n = 5; one-tail *t*-test, significant difference from the oHAP-BC group at the same HAP loading, * *p* < 0.05, ** *p* < 0.005).

**Figure 13 polymers-13-03860-f013:**
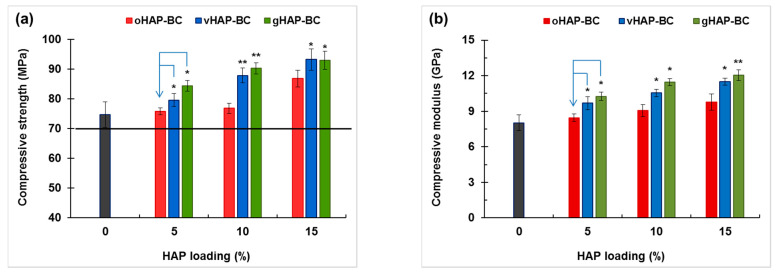
Compressive properties of HAP-modified bone cements as function of HAP loadings. (**a**): compressive strength and (**b**): compressive modulus. (Result = mean ± SD, n = 3; significant difference from one-tail *t*-test: * *p* < 0.05, ** *p* < 0.005, relative to oHAP-BCs at the same HAP loading).

**Figure 14 polymers-13-03860-f014:**
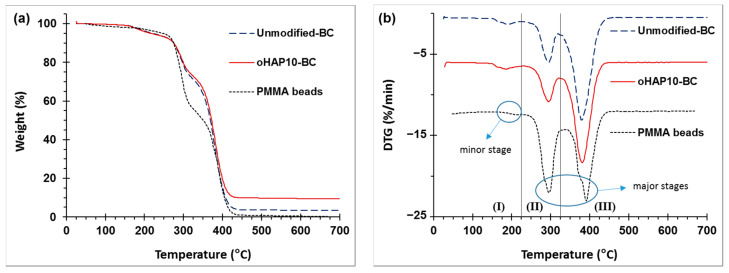
(**a**): TGA and (**b**): DTG curves of PMMA beads, unmodified BC and oHAP10-BC. Note: (unmodified-BC: 0 wt.% of HAP, with 5 wt.% of ZrO_2_).

**Figure 15 polymers-13-03860-f015:**
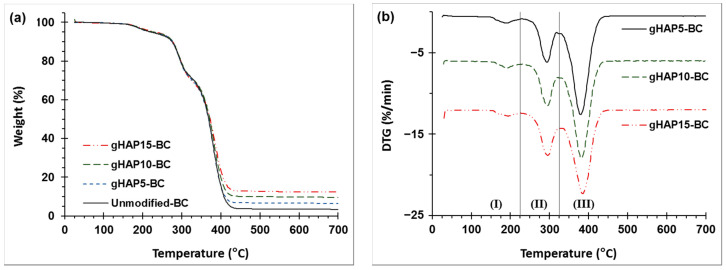
(**a**): TG and (**b**): DTG curves of PMMA bone cements modified with gHAP loadings of 0, 5, 10 and 15 wt.%. Note: (unmodified-BC: 0 wt.% of HAP, with 5 wt.% of ZrO_2_).

**Figure 16 polymers-13-03860-f016:**
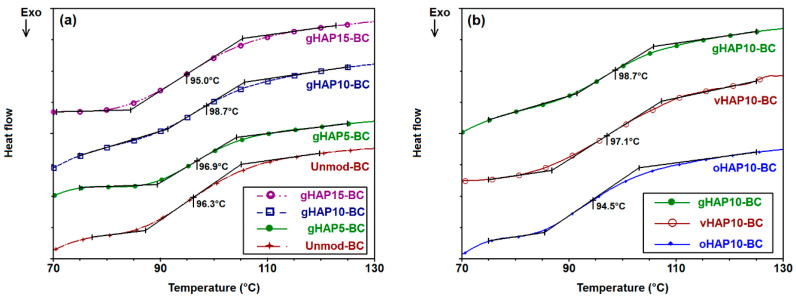
(**a**): DSC diagrams of gHAP-BC samples modified with gHAP loadings of 0, 5, 10, 15 wt.%, labeled as unmodified-BC, gHAP5-BC, gHAP10-BC and gHAP15-BC, respectively, and (**b**): DSC diagrams of bone cements modified with 10 wt.% of oHAP, vHAP and gHAP, labeled as oHAP10-BC, vHAP10-BC and gHAP10-BC, respectively.

**Table 1 polymers-13-03860-t001:** Composition of PMMA bone cements modified with HAPs.

Sample Name	Powder0 (g)	HAP (g)	vHAP (g)	gHAP (g)	Liquid (g)
Unmodified-BC	12.0	-	-	-	6.0
oHAP5-BC	12.0	0.6	-	-	6.3
oHAP10-BC	12.0	1.2	-	-	6.6
oHAP15-BC	12.0	1.8	-	-	6.9
vHAP5-BC	12.0	-	0.6	-	6.3
vHAP10-BC	12.0	-	1.2	-	6.6
vHAP15-BC	12.0	-	1.8	-	6.9
gHAP5-BC	12.0	-	-	0.6	6.3
gHAP10-BC	12.0	-	-	1.2	6.6
gHAP15-BC	12.0	-	-	1.8	6.9

**Table 2 polymers-13-03860-t002:** Setting properties of PMMA bone cements modified with three kinds of HAPs.

Property	Loading of oHAP (wt.%)				Loading of vHAP (wt.%)			Loading of gHAP (wt.%)		
	0	5	10	15	5	10	15	5	10	15
T_max_ ± 0.2 (°C)	64.9	64.0	62.2	61.5	63.5	61.8	61.1	63.2	61.4	60.8
T_set_ ± 0.2 (°C)	44.0	43.5	42.6	42.3	43.2	42.5	42.2	43.2	42.4	41.9
t_set_ ± 0.03 (min)	8.07	9.82	12.09	13.48	10.05	12.01	13.67	10.36	12.64	14.01
t_atmax_ ± 0.03 (min)	8.8	10.5	13.1	14.6	10.7	13.0	14.9	11.0	13.4	14.7
t_dough_ ± 0.1 (min)	3.75	3.75	4.0	4.0	3.75	3.75	4.0	3.75	4.0	4.0
t_work_ ± 0.13 (min)	4.3	6.1	8.1	9.5	6.3	8.3	9.7	6.6	8.6	10

**Table 3 polymers-13-03860-t003:** Bending properties of unmodified and modified bone cements modified 10 wt.% of HAPs before and after soaking in saline for 72 h.

Sample	Before Soaking		After Soaking		*p*-Value (n = 3, One-Tail *t*-Test)	
	BM (GPa)	BS (MPa)	BM (GPa)	BS (MPa)	BM (GPa)	BS (MPa)
Unmodified-BC	2.31 ± 0.05	65.51 ± 3.1	2.30 ± 0.04	64.87 ± 4.1	0.28 **	0.23 **
oHAP10-BC	2.18 ± 0.03	47.93 ± 1.3	2.15 ± 0.03	44.38 ± 1.3	0.10 **	0.02 *
vHAP10-BC	2.23 ± 0.05	52.71 ± 1.3	2.19 ± 0.07	51.28 ± 0.7	0.14 **	0.21 **
gHAP10-BC	2.27 ±0.04	59.30 ± 2.1	2.24 ± 0.07	57.16 ± 0.7	0.15 **	0.18 **

Note: Result = mean ± SD, n = 3; significant difference from one-tail *t*-test: between bending properties of HAP-BC sample before and after soaking; * indicates *p* < 0.05 (significant change), ** indicates *p* > 0.05 (no significant change).

**Table 4 polymers-13-03860-t004:** TGA results of unmodified bone cement and HAP-modified bone cements.

Sample	WL_1_ (%)	WL_2_ (%)	WL_3_ (%)	W_f_ Residue (%)
PMMA beads	3.94	40.10	55.69	0.30
Unmodified BC	5.29	23.49	67.89	3.36
gHAP5-BC	5.23	24.28	64.06	6.43
gHAP10-BC	4.89	23.73	61.80	9.59
gHAP15-BC	4.65	25.37	57.64	12.34
oHAP10-BC	4.73	25.30	60.56	9.41

## Data Availability

Data is contained within the article.

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
