# Peer review of "PMMA Bone Cements Modified with Silane-Treated and PMMA-Grafted Hydroxyapatite Nanocrystals: Preparation and Characterization"

_polymers, 2021, doi:10.3390/polym13223860_

Round 1

Reviewer 1 Report

I commend the authors for their research on enhancing the properties of bone cements by compositing widely used PMMA and HA bone cements. Attached is a word document detailing the suggestions I made. 

Author Response

Dear Reviewer 1,

Thank you for your suggestions and comments.

Below is our point by point response.

DQTham

Institute for Tropical Technology - VAST

[email protected]

Point 1: I would strongly suggest authors to perform the evaluations for soaked samples and add to the manuscript. If in case, this have been already tested and published, I would suggest adding few sentences or paragraphs citing the research showing that even on wetting the samples they retained their mechanical properties.

- We have already carried out the bending test of some selected samples after soaking in saline solution for 3 days at 25 °C (+/- 3) . The obtained results in new Table 3 and discussion were represent.

Point 2: Please mention the number of samples tested for each experiment. I noticed error bars in the graphs (Figure 11, 12, 13 and so on), but did not mention what the sample number is and what type of error bars were they. Please indicate those in the titles/annotations of the figures.

- We added the type of error bars, standard deviation (SD, n type), number of specimens (n), and asterisk marks for p-values as annotations in Figures 11, 12, 13.

Point 3: Similarly, in the tables (Table 1 and so on.), there are no deviations noted. I strongly suggest authors to include statistical parameters like n, p, type of error bars in the titles of the graphs and tables.

- Where appropriate, the results in Tables (2, 3) were expressed as mean +/- SD. (Results in Table 4 (new version) were calculated from TGA measurement, statistical analysis for these data is not necessary. As usual, we noted the parameters like n, p, type of error bars under graphs and tables.

Point 4: I would also suggest authors to add a statistical significance section to the manuscript describing all the statistical evaluations performed (like ANOVA or T-test) to prove the significance of the results obtained.

- We have add a statistical significance section (2.4) to the manuscript describing all the statistical evaluations performed (with T-test, parameters of “one-tail”, “pair for means”), significant change was verified with p < 0.05.

2.4. Statistical analysis

Data analysis was performed by using Microsoft Excel 2016. Where appropriate, results are expressed as mean ± standard deviation (SD, “n” method). T-test function with parameters of “one-tail” and “paired-sample for means” was used to analyze significant differences between a pair of bone cement samples. Significant difference was verified if p-values < 0.05.

REVIEWER1’s comments

I commend the authors for their research efforts to enhance the material properties of widely used PMMA bone cements. Hydroxyapatite’s osteoconductive and osteoinductive nature makes it a potential candidate for the replacement of PMMA bone cement, but its use is limited due to its lower mechanical strength. Authors tried to well integrate both the bone cements together to enhance the bone cements properties overall.

Introduction was to the point. Material and methods were sufficiently explained. All the results were well organized, and conclusions drawn from the results were logical.

Since the bone cements are usually applied in wet areas (as support materials or as fillers to the bones), there will be a continuous wet environment around the bone cements. As PMMA is a methacrylate polymer, wettability does not affect its properties. But, numerous previous researchers have shown that Hydroxyapatite properties may vary due to surrounding environment. It would have been wiser for the authors to test the properties of their novel PMMA-HA composite samples both dry and soaked in water/saline for a day. This way they could prove their composite’s mechanical strength in wet environment. I would strongly suggest authors to perform the evaluations for soaked samples and add to the manuscript. If in case, this have been already tested and published, I would suggest adding few sentences or paragraphs citing the research showing that even on wetting the samples they retained their mechanical properties.

In methods section, for few evaluations, it was mentioned that 5 samples were tested for each composition and for few experiments number of samples evaluated were not mentioned. Please mention the number of samples tested for each experiment. I noticed error bars in the graphs (Figure 11, 12, 13 and so on), but did not mention what the sample number is and what type of error bars were they. Please indicate those in the titles/annotations of the figures. Similarly, in the tables (Table 1 and so on.), there are no deviations noted. I strongly suggest authors to include statistical parameters like n, p, type of error bars in the titles of the graphs and tables. I would also suggest authors to add a statistical significance section to the manuscript describing all the statistical evaluations performed (like ANOVA or T-test) to prove the significance of the results obtained.

I would be great to see a future work section added to the manuscript too.

Reviewer 2 Report

Tham et al. reported the preparation and characterization of different hydroxyapatites, which were then used as additives to prepare different groups of hydroxyapatite-modified PMMA bone cements. This work is pretty comprehensive, and the manuscript is well organized. I believe that the  manuscript may attract the readership of Polymers. The only shortcoming is that the current manuscript lacks some discussions on the implications of the results emerged from this study. I would recommend the acceptance of this manuscript for publication once the authors have addressed this issue.

Author Response

Dear Reviewer 2,

Thank you for your suggestions and recommendation for publication of our research paper.

We have added some discussions on the implications of the results emerged from this study.

Below is our additions.

DQTham

Institute for Tropical Technology - VAST

[email protected]

Page 11

- In section 3.1, TG and FTIR analyses have demonstrated the presence of vinyl groups on surface of vHAP particles, which can be copolymerized with MMA to form PMMA-g-HAP hybrid, as well as the presence of grafted PMMA molecules on surface of gHAP particles (reaction scheme in Figure 2.c).

Page 14

- As mentioned in previous section 3.1, presence of vinyl groups and grafted PMMA molecules on surface of vHAP, gHAP nanoparticles, respectively, promotes the higher organic affinity of vHAP and gHAP nanoparticles, which improves their dispersion and interaction in the polymerized matrix better than original HAP nanoparticles (oHAP) does, thus improving tensile and bending strength.

Page 16

- Meanwhile, BS and BM of other samples (unmodified-BC, vHAP10-BC, gHAP10-BC) only show a slight decline trend after soaking. This decline trend could be due to the absorption of water under wet condition. FESEM images in Figure 7 shows that vHAP and gHAP particles are better dispersed and adhered with the matrix than oHAP particles, which may reduce the porosity and water absorption ability of bone cements.

Page 18

- There are two possible reasons for Tg of vHAP10-BC is higher than that of oHAP10-BC sample. The first is related to higher organic affinity of VTMS-grafted layer on surface of HAP nanoparticles and the second is the in situ formation of PMMA-g-HAP during curing process due to the copolymerization of vinyl groups of on surface of vHAP and MMA monomers, initialized by BPO and DMPT initiators. Meanwhile, interaction between oHAP particles and PMMA matrix is weaker than vHAP. It means that the mobility of PMMA chains in the vHAP10-BC system is lower, leading to its higher Tg in compared with oHAP10-BC.

-------

Reviewer 3 Report

My comments are in attached file.

Author Response

Dear Reviewer 3

Thank you for your suggestions, corrections and recommendation for publication of our research paper.

In overall, we have accepted all your corrections. The text is highlight in red. Moreover, we also added some more corrections. Below is your comments, corrections and our answers.

DQTham

Institute for Tropical Technology - VAST

[email protected]

Round 2

Reviewer 1 Report

I thank all the authors for considering my comments and suggestions. Manuscript looks amazing and I would recommend publishing it in this form.